# Hereditary Hypertrophic Cardiomyopathy in Children and Young Adults—The Value of Reevaluating and Expanding Gene Panel Analyses

**DOI:** 10.3390/genes11121472

**Published:** 2020-12-08

**Authors:** Eva Fernlund, Antheia Kissopoulou, Henrik Green, Jan-Erik Karlsson, Rada Ellegård, Hanna Klang Årstrand, Jon Jonasson, Cecilia Gunnarsson

**Affiliations:** 1Crown Princess Victoria Children’s Hospital, and Department of Biomedical and Clinical Sciences, Department of Pediatrics, Linköping University, 581 83 Linköping, Sweden; eva.fernlund@gmail.com; 2Department of Clinical Sciences Lund, Pediatric Heart Center, Skåne University Hospital, Lund University, 221 00 Lund, Sweden; 3Department of Internal Medicine, County Council of Jönköping, Department of Health, Medicine and Caring Sciences, Linköping University, 581 83 Linköping, Sweden; jan-erik.karlsson@rjl.se; 4Division of Drug Research, Department of Biomedical and Clinical Sciences, Linköping University, 581 83 Linköping, Sweden; henrik.green@liu.se; 5Department of Forensic Genetics and Forensic Toxicology, National Board of Forensic Medicine, 111 64 Stockholm, Sweden; 6Department of Clinical Genetics, and Department of Biomedical and Clinical Sciences, Linköping University, 581 83 Linköping, Sweden; Rada.Ellegard@regionostergotland.se (R.E.); Hanna.Klang.Arstrand@regionostergotland.se (H.K.Å.); ja.jonasson@icloud.com (J.J.); Cecilia.Gunnarsson@regionostergotland.se (C.G.); 7Centre for Rare Diseases in South East Region of Sweden, Linköping University, 581 83 Linköping, Sweden

**Keywords:** pediatric cardiomyopathy, hypertrophic, exome sequencing, gene panel, sudden cardiac death

## Abstract

Introduction: Sudden cardiac death (SCD) and early onset cardiomyopathy (CM) in the young will always lead to suspicion of an underlying genetic disorder. Incited by the rapid advances in genetic testing for disease we have revisited families, which previously tested “gene-negative” for familial predominantly pediatric CM, in hopes of finding a causative gene variant. Methods: 10 different families with non-syndromic pediatric CM or hypertrophic cardiomyopathy (HCM) with severe disease progression and/or heredity for HCM/CM related SCD with “gene-negative” results were included. The index patient underwent genetic testing with a recently updated gene panel for CM and SCD. In case of failure to detect a pathogenic variant in a relevant gene, the index patient and both parents underwent clinical (i.e., partial) exome sequencing (trio-exome) in order to catch pathogenic variants linked to the disease in genes that were not included in the CM panel. Results: The mean age at clinical presentation of the 10 index cases was 12.5 years (boys 13.4 years, *n* = 8; girls 9 years, *n* = 2) and the family history burden was 33 HCM/CM cases including 9 HCM-related SCD and one heart transplantation. In 5 (50%) families we identified a genetic variant classified as pathogenic or likely pathogenic, in accordance with the American College of Medical Genetics and Genomics (ACMG) criteria, in *MYH7* (*n* = 2), *RBM20*, *ALPK3*, and *PGM1*, respectively, and genetic variants of unknown significance (VUS) segregating with the disease in an additional 3 (30%) families, in *MYBPC3*, *ABCC9*, and *FLNC*, respectively. Conclusion: Our results show the importance of renewed thorough clinical assessment and the necessity to challenge previous genetic test results with more comprehensive updated gene panels or exome sequencing if the initial test failed to identify a causative gene for early onset CM or SCD in children. In pediatric cardiomyopathy cases when the gene panel still fails to detect a causative variant, a trio exome sequencing strategy might resolve some unexplained cases, especially if a multisystemic condition is clinically missed.

## 1. Introduction

Pediatric cardiomyopathy has an annual incidence of 1.1–1.5 per 100,000 children under the age of 18 years [1,2,3]. It has the same etiology as cardiomyopathy (CM) in adults, but especially in infants the CM can be part of a syndromic, neuromuscular, or metabolic disease [2,3,4,5,6,7,8,9], and the prognosis is dependent on the underlying condition. In children, Hypertrophic cardiomyopathy (HCM), Dilated cardiomyopathy (DCM), Restricted cardiomyopathy (RCM), and Left ventricular non-compaction cardiomyopathy (LVNC) can occur isolated or as a mixed cardiomyopathy (a phenotypic overlap) that can complicate a more specific categorization. CM is usually inherited in an autosomal dominant pattern, sometimes with reduced penetrance, and is most often caused by variants in genes encoding components of the sarcomere, cytoskeleton, or desmosome [10,11].

HCM is the most common monogenic cardiac disorder and it is a leading cause of sudden cardiac death (SCD) in the young. It is characterized by disturbed myocardial architecture involving intracellular, extracellular, peri- and microvascular changes [12,13,14,15], gradual thickening of the myocardium and an increased risk for major cardiac events, including ventricular arrhythmias and SCD [16,17,18].

Genetic testing is highly recommended in clinical cases of familial or pediatric HCM, because an actionable disease-causing genetic variant can be found in nearly two thirds of the patients [6,10,17,19,20,21]. The corresponding figure for HCM in general is obviously lower. The benefit of presymptomatic genetic diagnosis is that high-risk individuals at any age can be included in clinical follow-up programs with possibilities for early intervention, while the non-carriers can be exempted from further clinical investigation and follow up.

For families with hereditary cardiomyopathy where a causative genetic variant has not been identified, i.e., “gene-negative” HCM, genetic cascade-screening is not possible.

The aim of the current study was to identify the missing genetic etiology of the “gene-negative” cases in a cohort of pediatric CM, mainly early onset HCM, where previously performed clinical genetic testing had been unable to find a causative genetic variant.

## 2. Materials and Methods

In previous studies of a Swedish cohort [22,23] with either pediatric and early onset HCM, pediatric HCM with severe disease progression, heredity for HCM-SCD or where the index patient suffered SCD due to HCM, the clinical genetic testing revealed pathological results in 40 of 54 young HCM-patients (74%) from 46 unrelated families. Disease causing genetic variants were found mainly in *MYBPC3* and *MYH7,* but also in *TNNT2, TNNI3, TCAP*, and *PRKAG2*. The participants in the current study are “gene-negative” pediatric CM patients (mainly HCM), recruited from this former study (6 patients) and 4 patients with clinical suspicion of familial cardiomyopathy from the South and Southeast healthcare regions in Sweden who met the inclusion criteria below.

### 2.1. Inclusion Criteria

The pediatric index patients who fulfilled clinical criteria (ESC) regarding HCM <18 years old and/or had a family history of heart transplantation (Htx) or SCD due to HCM/CM before the age of 45 years were eligible for inclusion in the present study if previous clinical genetic screening had failed to demonstrate a causative genetic variant. The parents of the index patients and additional cases of clinical HCM/CM in the family were also invited to participate in the study. All participants or their parents signed a written informed consent to participate in this study. Patients with left ventricular hypertrophy (LVH) as part of a syndromic diagnosis were excluded from the study.

### 2.2. Methods

DNA extraction from whole blood samples was performed using either EZ1 (Oiagen) or Prepito (Techtum). DNA concentration and quality were determined using a NanoDrop spectrophotometer. Samples with A260/A280 ratios above 1.8 and A260/A230 above 1.5 were accepted for further analysis. NGS libraries were prepared using TruSight One Expanded sequencing panel (Illumina, Eindhoven, The Netherlands), which contains 6794 genes with known clinical significance and sequenced using a NextSeq 500 instrument (Illumina, Eindhoven, The Netherlands). The coding region and the flanking exon/intron boundaries of 60 genes associated with heart function (Table 1) were analyzed using an in-house pipeline in CLC Biomedical Genomics Workbench (Qiagen, Sollentuna, Sweden) with hg19 (GRCh37.p10 Primary Assembly) as a reference. The virtual 60 gene panel design was largely based on the relevant Genomics England PanelApp “green genes” [24]. Variants were assessed according to American College of Medical Genetics and Genomics [25] (ACMG) guidelines including deleteriousness prediction scoring methods (SIFT, Mutation Taster, PolyPhen-2, CADD) for variant interpretation. The identification of candidate gene variants in the trio-exome analysis was performed using filtering criteria tailored according to each patient’s phenotype and presumed inheritance pattern (autosomal dominant, autosomal recessive, or de novo) using Ingenuity Variant Analysis software (Qiagen, Sollentuna, Sweden). Genes of interest were selected according to custom gene lists based on published reports on gene-phenotype correlations and/or by filtering and ranking the variants using phenotype-driven ranking in Ingenuity, the work-flow is summarized in Figure 1.

All index patients had previously undergone genetic testing on a clinical basis, but the genetic test used depended on the referring clinic, analyzing laboratory and the year of the analysis. The genetic analysis of the first included 6 families was performed by DNA sequencing and examination of the coding exons of *MYH7, MYBPC3, MYL2, MYL3, TNNT2, TPM1, ACTC, TNNI3, CSRP3, TCAP*, and *PLN.* Later on, additional genes have been added to the panel. Of note, the “old” genetic test did not have full coverage of all coding regions of the genes and only the variants considered pathogenic or possibly pathogenic at the time of analysis were reported. To adjust for these differences, index patients were first re-analyzed as described above using a new blood sample. In case a “pathogenic” or “likely pathogenic” variant was not found with our virtual 60 gene panel, the sample was reanalyzed together with samples from both parents as a “Trio-exome” using the full sequencing panel of 6794 clinical exome genes. As described above, trio analysis was performed by filtering the variants according to the suspected inheritance pattern, in some cases followed by phenotype-driven ranking of the resulting variants according to relevant human phenotype ontology (HPO) terms using Ingenuity Variant Analysis software (Qiagen, Sollentuna, Sweden). For confirmation of mutation status in family members, Sanger sequencing was performed using BigDye Terminator v.3.1 Cycle Sequencing Kit on an AB 3500xL (Thermo Fisher Scientific, Stockholm, Sweden).

## 3. Results

Ten families fulfilled the inclusion criteria and were invited to the present study of patients where the initial clinical genetic screening had failed to find a causative genetic variant. Among the included families 3 of the index patients had a mixed type of CM (HCM/RCM, Family 6; HCM with influence of LVNC, Family 3; and a familial burden of LVNC and DCM, Family 9).

The 10 familial CM cases had mean age 12.5 years at presentation, with a family burden of 33 HCM/CM cases including 9 cases of HCM-related SCD and one heart transplantation (Htx) (Table 2). At the time of this study there were 67 family members at risk for CM. The pedigrees of the included families are displayed in Figure 2. The results of the genetic analyses in these families are described below and summarized in Table 2, and the assessment of the pathogenicity of the specific genetic variants is shown in Table 3.

### 3.1. “Pathogenic”/”Likely Pathogenic” Genetic Variants

**Family 1**: The index patient (II:1) in this family was a 17-year-old girl with HCM phenotype since age 7, asymptomatic, but found to have myocardial hypertrophy at the initial examination because of heredity for HCM. During follow-up this girl had an accelerated myocardial growth (interventricular septum (IVS): 18 mm, +4.7SD, Detroit Z-score) [26], high risk by ECG-risk score [27] analysis (ECG and echocardiogram, Figure 3), and myocardial ischemia in the exercise stress test. Due to this alarming progression of the disease, she had an ICD at the age of 12 years. Her father (I:1) had a heart transplant because of HCM at age of 47 years and her uncle (I:2) suffered SCD due to HCM at the age of 49 years. The initial genetic analysis (of the father of the index patient) had been performed in 2007 with a “normal” result. The present CM panel performed on the index patient revealed a previously described “pathogenic” heterozygous variant in *MYH7* NM_000257.3 c.746 G>A; (p.R249Q): This genetic variant segregated with HCM in this family. The variant was clinically reported as pathogenic according to ACMGs guidelines [28,29]. The father (I:1) of the index patient had previously been tested with a “normal” result by an external laboratory using Sanger sequencing and a panel with the following genes: *MYH7, MYBPC3, MYL2, MYL3, TNNT2, TPM1, ACTC, TNNI3, CSRP3, TCAP*, and *PLN*. The reason for the absence of a positive finding remains unknown.

**Family 8:** The index patients in this family are two siblings III: 1 and III: 2 with a family history of an inherited lethal cardiac disease in the diseased mother (II:1) and the grandmother (I:1). Prior genetic testing of the siblings was performed by the initial clinic in Syria, and reported to be normal. Documentation of laboratory tests missing. The 12-year-old boy (III: 1) was found to have severe HCM by echocardiogram (IVS 21 mm (+5.4SD), posterior wall (PW) 11 mm (3.2SD)). Using the CM panel, we found a previously described heterozygous “likely pathogenic” variant in *MYH7* NM_000257.3 c.1063 G>A (p.A355T) [28,30]. The sister (III: 2) had normal ECG and echocardiogram, and did not carry the genetic variant and could be released from further follow-up.

**Family 9:** The index patient is a 15-year-old boy who presented with recurrent palpitations. The ECG showed: sinus rhythm without preexcitation, 55 beats/min, QRS-duration 114 ms, intraventricular block without signs of hypertrophy. The initial echocardiogram showed left ventricular end-diastolic diameter measured at 57 mm (z-score +3SD), normal IVS: 8 mm with generally impaired contractility, ejection fraction (EF) 40%. The boy has a 13 year-old brother (III: 2) who has experienced breathing problems. The echocardiogram showed left ventricular non-compaction cardiomyopathy (LVNC) with a moderately enlarged left ventricle with preserved systolic function. Furthermore, the patient’s mother (II: 1) presented with chest pain and pre-syncope symptoms. Echocardiogram showed mild systolic left ventricular dysfunction and excessive trabeculations on the apical wall. The boy’s grandmother (I: 1) died at the age of 68, due to heart problems. The boy’s aunt (II:2) has at the age of 47 developed heart failure, where the echocardiogram showed a moderately dilated left ventricle with EF 30–35% and signs of hyper-trabeculation.

Genetic analysis with the NGS CM-panel showed a heterozygous “likely pathogenic” variant in *RBM20* NM_001134363.2 c.1912C>G (p.P638A), previously described by Brauch et al. [31], but also a heterozygous variant of unknown significance (VUS) in the *VCL* gene c.2969 C>T (p.A990V). It turned out that the brother, the mother, and the aunt are also carriers of the familial mutation *RBM20* but the *VCL* variant has not been detected in either of them. Previous genetic testing (HCM gene panel, which did not include *RBM20* or *VCL*) of (II: 2) performed by an external laboratory did not reveal any pathogenic variants.

### 3.2. Pathogenic/Likely Pathogenic Genetic Variants Found by Trio-Exome in Selected Cases

**Family 6:** The index patient had short stature and cleft palate. Due to the clinical picture, an echocardiographic examination was performed that revealed mild left ventricular hypertrophy of restrictive pattern and enlarged atrias. The index patient died suddenly at the age of 13 during physical activity. The clinical phenotype was shared with two siblings, one of these died at 12 years of age after a short rush up the stairs at home. A homozygous variant NM_002633.2:c.689 G>A in *PGM1* (p.G230E), known to cause congenital disorders of glycosylation (*PGM1*-CDG) in homozygous carriers was found in this family after phenotype-driven ranking of variants according to HPO terms. This family was described in detail in a previous publication [32]. The previously negative test had been performed by an external laboratory with a panel including the following genes: *MYH7, MYBPC3, MYL2, MYL3, TNNT2, TPM1, ACTC, TNNI3, CSRP3, TCAP*, and *PLN,* and also by our own laboratory using the CM 60 gene panel.

**Family 10:** The index patient fainted during sports activity at 14 years of age. It was attributed to a vasovagal syncope episode, totally missing the obvious signs of LVH on the ECG (Figure 4). He presented at age 24 with renal disease. Perioperative ECG changes were observed on the monitor that led to further cardiological assessment. The echocardiogram revealed myocardial hypertrophy, mainly comprising the midventricular septum, measuring 28 mm with an LVEF of 50% without any signs of outflow obstruction. In the family history, his mother had HCM diagnosed incidentally at age 42. The brother of the maternal grandfather died suddenly at the age of 56 without previous cardiac symptoms. The grandfather was clinically screened at age 54, where the echocardiogram depicted a severely hypertrophied septum up to 30 mm. Holter revealed several episodes with non-sustained VT, leading to an ICD implantation. Our index patient is also planned to receive an ICD as he has had several episodes of non-sustained VT on the holter monitor. The mother underwent the CM 60 gene panel test in 2015 and our index in 2019 without finding any pathogenic variants. Trio-exome unraveled the likely cause of the HCM in this family when a heterozygous variant in *ALPK3* (Chr15:85370829 NM_020778.4:c.903delC) gene was detected [33] considered as “likely pathogenic” according to ACMGs guidelines. This variant was found in our index, his mother, and his grandfather, segregating with the disease.

### 3.3. Genetic Variants of Unknown Significance (VUS)

**Family 3:** The index patient (II: 1) presented at age 14 with palpitations triggered by exercise. ECG showed prolonged QTc, inferior ST-T changes (Figure 5) and echocardiogram only mild LVH (IVS 14 mm, +2.8SD). During puberty there was a rapid progression of myocardial hypertrophy (at 17 years IVS 23 mm, (+4.4SD) measured by echocardiogram and CMR. By family clinical screening the mother (I: 1) was found to have a muscular VSD accompanied by left ventricular non-compaction cardiomyopathy. Trio-exome showed a variant of unknown significance (VUS) in the gene *ABCC9*; heterozygous Chr12:21997457 NM_005691.3 c.3275T>G (p.I1092S) present in both the index patient and the mother. This genetic variant of unknown significance is found in 0.006% in the normal population and had been assigned as “not reportable” in the preceding CM panel analysis. Genetic variants in *ABCC9* have previously been described to cause DCM. ClinGen reevaluation suggests there is little evidence to support this association (https://search.clinicalgenome.org/kb/genes/HGNC:60).

**Family 4**: The index patient (II: 4) presented at 5 years of age due to heredity of HCM, where the uncle (I: 3) was deceased in HCM. The boy had no symptoms and a normal echocardiogram, but an ECG with complete right bundle branch block (RBBB) at first visit. During follow-up there was a progression to clinical HCM with asymmetric interventricular septum (IVS > 2SD) by echocardiogram. The younger sister (II: 1) was found to have a mild HCM (IVS and PW 6 mm, +2.5SD at 6 months of age), the father (I: 1) as well as the other uncle (I: 2) had clinical HCM. Trio-exome sequencing showed a heterozygous genetic variant of unknown significance (VUS) in the *FLNC* gene [34], NM_001458.4 FLNC c.7559 C>T, (p.T2520I) found to segregate with clinical disease in this family. The previously negative test performed by an external laboratory had included the following genes: *MYH7, MYBPC3, MYL2, MYL3, TNNT2, TPM1, ACTC, TNNI3, CSRP3, TCAP*, and *PLN.*

**Family 5**: Index patient (II:1) in this family was a boy with HCM with a murmur due to a very severe left ventricular outflow tract obstruction (LVOTO, Vmax 5.2 m/s, gradient > 108 mmHg) diagnosed in the first days in life. The echocardiogram (Figure 6A,B) showed there was a rich amount of mobile accessory material in the aortic subvalvular area, mainly localized in the anterior leaflet of the mitral valve, and a massive asymmetric septal hypertrophy (IVS 17 mm (+6.9SD), PW 4.4 mm (+1.8SD)). A neonatal heart operation was performed by myomectomy ad modum Morrow including restoration of the LVOT area. The father (I: 1) also had neonatal onset of HCM but without the need of heart surgery. In this family we found a heterozygous variant in *MYBPC3* NM_000256.3 c.2320G>A (p.A774T). This variant has previously been reported [35] in cases of CM but was assessed as a variant of unknown significance (VUS) according to ACMG guidelines. The previously negative test performed by an external laboratory included the following genes: *MYH7, MYBPC3, MYL2, MYL3, TNNT2, TPM1, ACTC, TNNI3, CSRP3, TCAP*, and *PLN.* The reason for not reporting the *MYBPC3* variant remains unknown.

### 3.4. Families without Genetic Findings by Current Strategy

**Family 2:** The proband in this family (I: 1) who suffered SCD at the age of 42 years was the mother of 6 children. Echocardiogram showed mild LVH with apical involvement. A totally normal ECG was performed two years prior to the lethal cardiac event. Postmortem autopsy confirmed the HCM diagnosis. Of the 6 children 2 of the boys (II: 3 and II: 5) had an echocardiographic picture of mild apical hypertrophy, IVS 14 mm (+3.1SD) but no clinical symptoms. The present strategy and genetic testing using the current CM panel and clinical exome did not reveal any genetic variant associated with HCM or SCD.

**Family 7:** The proband (II: 1) is a young boy with aortic stenosis that has been followed from early childhood. At the age of 12 years there was a rapid and progressive myocardial hypertrophy, mainly asymmetric septal hypertrophy (IVS 16 mm (+3.1SD), PW 11 mm (+1.9SD), and evolvement to a moderate LVOTO. He shared the echocardiographic feature of asymmetric septal hypertrophy with his father (I: 1), but the father had no LVOTO. Trio-exome screening did not reveal any genetic variant that could be connected to HCM.

## 4. Discussion

Previous studies have shown that pediatric cardiomyopathy frequently has a genetic background. However, in some patients the clinical genetic test fails to detect a causative genetic variant even though the patient has a severe cardiomyopathy and presents with an obvious family history of cardiomyopathy.

In the present study, incited by the rapid advances in genetic testing for disease we have revisited families, which previously tested negative for non-syndromic pediatric CM, in hopes of finding a causative gene variant. Index patients from 10 different families who had pediatric HCM/CM with severe disease progression and/or heredity for HCM/CM related SCD with “gene-negative” results of the clinical genetic testing were included.

The negative tests of the probands in these families had been performed by different commercial and hospital laboratories (including our own) with different gene-panels and sequencing technologies over a period of a few years. Accordingly, we felt that there was a need for more standardized analyses. In addition, the methods of interpretation had changed considerably over time. No doubt, knowledge of the genetics of CM has increased in recent years. Therefore, we thought a suitable strategy would be to re-analyze these families using our current method i.e., all sequencing is performed as a clinical exome sequencing run, the data of which is analyzed with virtual ad hoc gene panels; even single gene analysis is allowed. In this case, we started the analysis with an updated CM/SCD virtual gene panel, which included 60 genes with known association to CM or SCD (Table 1). In case a pathogenic or likely pathogenic variant was not found, the analysis was complemented by trio-exome sequencing analysis, including parental DNAs and analysis with an ad hoc gene panel based on further clinical assessment of the patient, which was interpreted by phenotype driven ranking (PDR). Thus, filtering criteria were tailored according to each patient’s phenotype and presumed inheritance pattern (autosomal dominant, autosomal recessive, or de novo) using Ingenuity Variant Analysis software (Qiagen, Sollentuna, Sweden).

In 5 out of 10 families we detected genetic variants that could be linked to CM and interpreted as pathogenic or likely pathogenic using ACMG criteria. In a further 3 families, variants of unknown significance (VUS) in CM associated genes were found segregating with the disease, and 2 families remained “gene-negative”, even after clinical exome sequencing using DNA from the index patient and the parents.

Das et al. (2014) investigated probands with HCM who earlier had been found to have a pathogenic variant or variant of unknown significance (VUS); each variant was reassessed by a panel of four reviewers and 10% of the genetic variants were reclassified to another group using ACMGs criteria [36]. These results are in line with our findings that a subgroup of patients with a negative result on the initially performed genetic testing were found to have a pathogenic or likely pathogenic variant when using an updated NGS panel. Although this study focuses mainly on pediatric HCM, many of the issues discussed are also relevant to other inherited cardiomyopathies as shown in family 9 with a more dilated and non-compaction left ventricular phenotype. Our findings suggest that perseverance and a repeat diagnostic testing using updated gene panels can be recommended if the initial genetic test failed to identify a causative gene variant in early onset CM or unexpected cardiac death in the young, especially if there is a family history of CM.

We also conclude that a trio-exome sequencing strategy cannot be expected to contribute much to the understanding of non-syndromic HCM cases or SCD due to HCM/CM in pediatric patients, as compared with an appropriate gene panel. Cirino et al. compared the results from NGS panels in 41 HCM patients with whole genome sequencing (WGS) and concluded that WGS and panel testing provided similar diagnostics yield in adult patients [37]. In contrast, Rupp et al. studying young patients with diagnosis of HCM before 18 years concluded that WES/WGS was an alternative after negative panel diagnostics [38]. Rupp et al. pointed out that panels analyzing genes associated with CM should also focus on multisystemic conditions when applied in younger children. In our study, we excluded patients with a known syndromic cause to CM, such as e.g., rasopathy or chromosomal disorder. Nevertheless, in family 6, by clinical reassessment and trio-exome sequencing we found a homozygous variant in the gene for phosphoglucomutase 1 (*PGM1*), not included in the CM gene panel (Table 1). This variant is associated with a congenital disorder of glycosylation (CDG), which has an autosomal recessive mode of inheritance [39]. Thus, the relatively small contribution of trio-exome sequencing to the diagnostic yield in our study as compared with the CM gene panel is explained by the exclusion of patients with a known syndromic cause to the CM. The genes used in the trio-exome sequencing to detect variants associated with CDG, are shown in Table 4. In complex syndromic conditions characterized by CM in children, the usage of trio exome as a first-tier diagnostic tool can translate into a decrease in medical interventions and incremental savings.

Additionally, Bagnall et al. pointed out that WGS improves outcomes of genetic testing in patients with severe HCM < 18 years of age with gene-elusive HCM [40]. They identified a clinically relevant variant in 9 of 46 (20%) families with gene-elusive HCM, and extending the genetic screening to intronic regions identified a further variant in 4 of 46 (9%) families.

In family 10, the trio exome analysis revealed a heterozygous variant in the *ALPK3* gene in all family members with severe hypertrophy. *ALPK3,* was first described only for pediatric recessive CM caused by homozygous truncating mutations [41]. There were only 2 of 10 heterozygous family members in previously described cases that showed hypertrophy and with an atypical distribution of hypertrophy [33]. Additionally, in a recent study [42], the first case of symptomatic *ALPK3*-associated cardiomyopathy presenting in adulthood is reported, leading to cardiac transplantation 10 years after the presentation of symptoms. This patient, though, carried a homozygous nonsynonymous single-nucleotide variant in exon 10 of *ALPK3,* underscoring the variability of symptoms and onset of disease that is seen in cases of cardiomyopathy caused by biallelic variants in *ALPK3.* On the contrary, Cheawsamoot et al. [43] have very recently reported that a monoallelic *ALPK3* variant can cause adult-onset HCM with a penetrance of 70% in a Thai family with 18 individuals, seven of whom had an adult onset HCM. In all seven affected and three unaffected members a heterozygous variant in *ALPK3* was identified. Our study also suggests that heterozygous variants of *ALPK3* (which was not included in the CM panel) can cause autosomal dominant HCM. Accordingly, *ALPK3* ought to be included in HCM gene panels.

In our cohort, three patients had variants identified in genes (*ABCC9*, *FLNC*, *MYBPC3*) that have associations with different cardiomyopathies and may contribute to the clinical phenotype in these families. These variants segregate with disease in the specific family but are determined as VUS using the current clinical guidelines. The uncertainties of variant interpretation have been a key limiting factor for clinical application of genetic testing. The ACMG rules for classification are supposed to minimize the risk for false positive interpretations, which can cause harm to the patients and their families. However, it is also well recognized that these stringent criteria result in under-calling of pathogenic variants in well-established cardiomyopathy genes [44]. The likelihood of obtaining a positive genetic test result is often dependent on whether the putative causative variant has been previously described and characterized. Walsh et al. (2019) [44] have shown that a disease and gene-specific approach with quantitative statistical methods can increase the accuracy and sensitivity of predictive genetic testing in HCM. A ClinGen cardiomyopathy variant curation expert panel is currently trying to adapt the ACMG classification criteria for genes associated with inherited cardiomyopathy. In this context, Kelly et al. [45] have presented adapted rules to provide increased specificity in *MYH7*-associated disorders. The application of quantitative variant classification methods to all clinically important genes remains a formidable task.

With the current strategy we failed to detect the underlying genetics in two families. In the apical form of HCM as in family 2, it has previously been described that it is less likely to find the causative underlying genetics [46,47]. HCM is a global disease [48,49] but the genetic evaluation mainly stems so far from research on western world populations and we have minor knowledge of the spectra of genetic variants of significance in some other parts of the world.

The reason why some of the variants that we detected had been missed at the first testing remains largely unknown. Surprisingly, 2 variants in the *MYH7* gene and 1 variant in the *MYBPC3* gene were identified. These genes are the two most commonly mutated in HCM gene panels. Possible reasons for missing these variants could be e.g., lack of coverage, other variant interpretation criteria, etc.

In conclusion, our results show the importance of renewed thorough clinical assessment and the necessity to challenge previous genetic test results with more comprehensive updated gene panels if the initial test failed to identify a causative gene for early onset CM or SCD in children. In pediatric cardiomyopathy cases when the gene panel still fails to detect a causative variant, a trio exome sequencing strategy might resolve some unexplained cases, especially if a multisystemic condition is clinically missed.

### Limitations of Our Study

This study describes the re-analysis of a small case series of 10 gene-negative HCM families as a proxy for the negative findings in a larger study where the diagnostic yield in the original cohort was 74%, showing that a genetic background can be identified in most cases of CM in children. Even if some of the “pathogenic” variants in the original cohort may represent false positives, it is outside the scope of this study to re-analyze them with contemporary ACMG guidelines. Although very limited in size, which might entail a risk of overinterpretation, the present study indicates a genetic cause for nearly all pediatric hypertrophic cardiomyopathy. Most of the initial testing that came out negative was performed by an external commercial laboratory using Sanger sequencing and a panel with the following 11 genes: *MYH7, MYBPC3, MYL2, MYL3, TNNT2, TPM1, ACTC, TNNI3, CSRP3, TCAP*, and *PLN*. The test did not include genes for metabolic HCM in children, such as *LAMP2* for Danon’s disease, *PRKAG2* etc., and produced some unreliable results for which we have received no explanation. Despite those caveats, the trials and tribulations of variant interpretation using the ACMG guidelines for classification became evident. This raises the important question of which genes, or parts thereof, ought to be included in a pediatric CM panel. Hopefully, a disease and gene-specific approach with quantitative statistical methods for interpretation can help our understanding of which variants are bad and which ones are good, and thus increase the accuracy and sensitivity of predictive genetic testing in HCM. Future studies with larger number of pediatric CM samples must confirm the results of this work.

## Figures and Tables

**Figure 1 genes-11-01472-f001:**
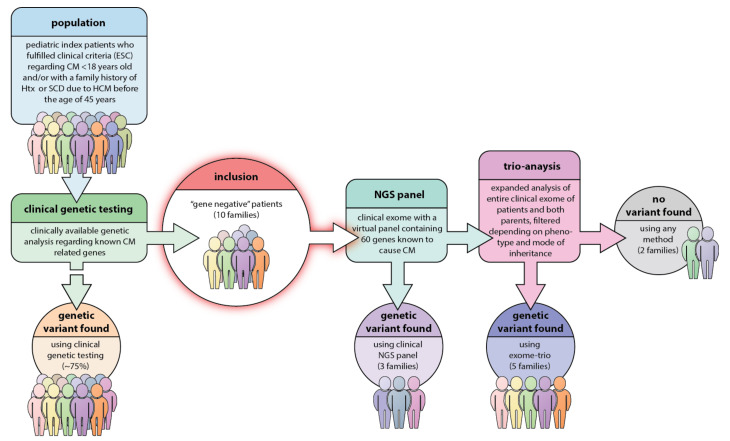
Study design and work-flow in our study. In the original young hypertrophic cardiomyopathy (HCM)-population, nearly 75% were found to carry a previous known HCM-causative genetic mutation. With the present strategy there was a result of a genetic variant interpreted as pathogenic in 5 families, i.e., 50% of the population and genetic variants of unknown significance (VUS) in additional 30%. Only in two families did this strategy fail to find the underlying genetics.

**Figure 2 genes-11-01472-f002:**
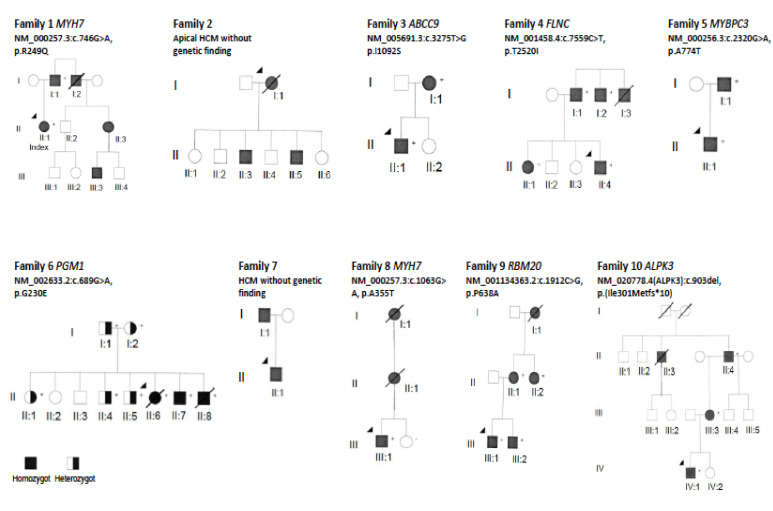
Pedigree of the families in this study. Circles in the pedigree denote females, squares males. A crossed-over symbol indicates that the particular individual has died. The arrow points out the index patient in each family. The autosomal dominant genetic variant detected in each family is indicated with a ‘+’ sign if the individual is a carrier. The black color filling indicates clinical disease presentation. Of note, in Family 6 there is the autosomal recessive disorder *PGM1*-CDG, where the black color filling show homozygous patients with clinical disease, and half-filling represent heterozygous asymptomatic carriers.

**Figure 3 genes-11-01472-f003:**
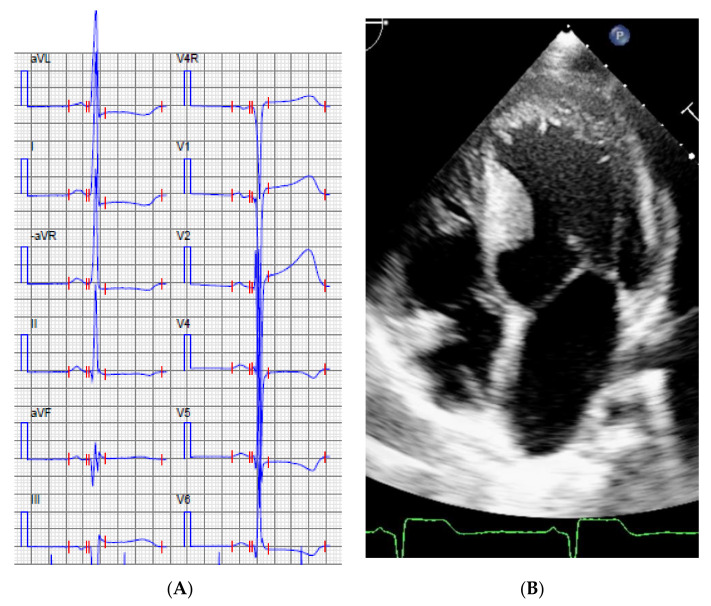
Index patient, Family 1, ECG (**A**) and the echocardiogram, apical 4-chamber view (**B**) at the age of 12 years.

**Figure 4 genes-11-01472-f004:**
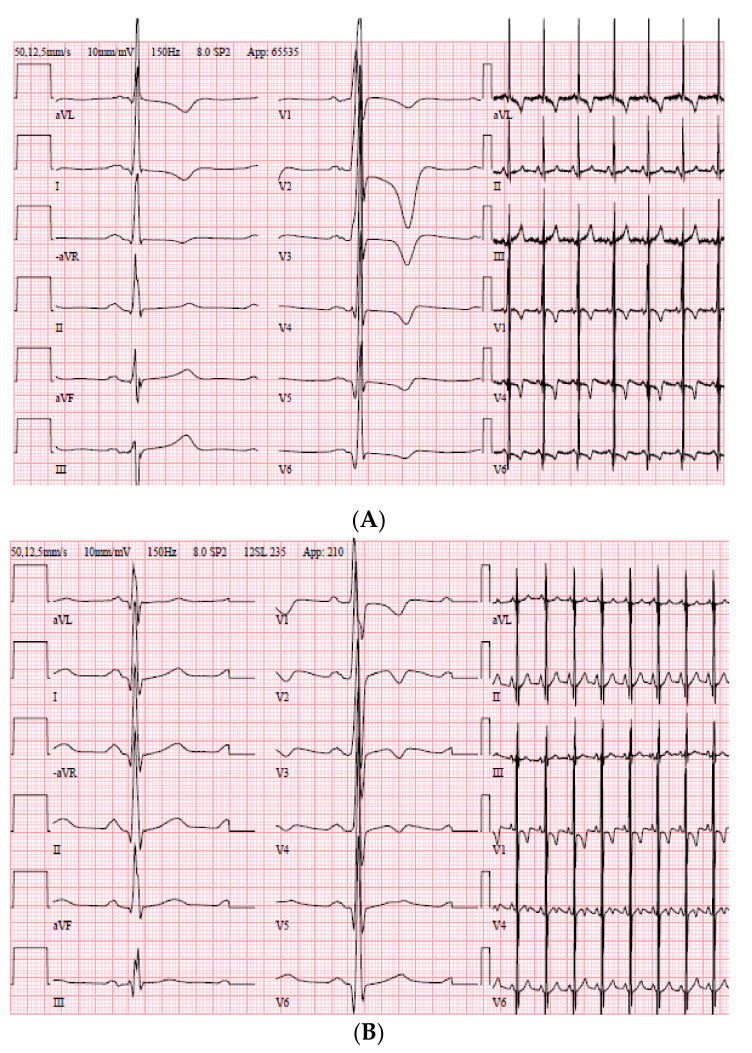
ECG of the index patient, Family 10. (**A**) ECG at the age of 24 years. (**B**) ECG at the age of 14 years when visited the emergency department because of a fainting episode after playing hockey.

**Figure 5 genes-11-01472-f005:**
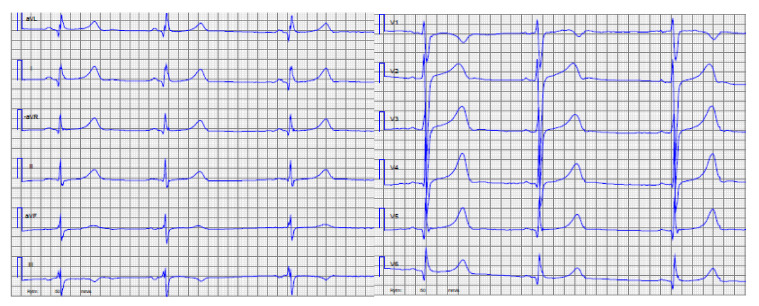
Index patient, Family 3, ECG at the age of 14 years with prolonged QTc. At this time, the echocardiogram revealed only mild left ventricular hypertrophy.

**Figure 6 genes-11-01472-f006:**
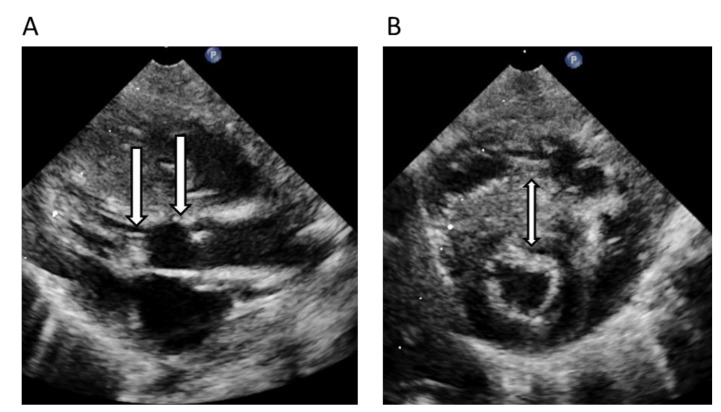
Echocardiogram preoperative in neonatal period of the index-patient, Family 5, left ventricular outflow tract obstruction (LVOTO) and left ventricular hypertrophy (LVH) in long axis view (**A**) and short axis view (**B**). Arrow located in the left ventricular outflow tract with accessory material creating a severe LVOTO (**A**), arrow also at the level of the aortic valve, while arrow in (**B**) are localized in the hypertrophied interventricular septum.

**Table 1 genes-11-01472-t001:** Genes associated with cardiomyopathy used in the cardiomyopathy (CM) 60 gene-panel.

*ABCC9*	*DSG2*	*HFE2*	*MYH7*	*PSEN1*	*TCAP*
*ACTC1*	*DSP*	*IDH2*	*MYL2*	*PSEN2*	*TFR2*
*ACTN2*	*EPG5*	*JUP*	*MYL3*	*RAB3GAP2*	*TNNC1*
*ANKRD1*	*EYA4*	*JPH2*	*MYLK2*	*RYR2*	*TNNI3*
*BAG3*	*FHL1*	*LAMP2*	*MYPN*	*RBM20*	*TNNT2*
*CSRP3*	*FKTN*	*LDB3*	*NEXN*	*SCN1B*	*TMEM43*
*CRYAB*	*GATAD1*	*LMNA*	*PKP2*	*SCN5A*	*TPM1*
*DES*	*GLA*	*LZTR1*	*PLN*	*SGCD*	*TTN*
*DMD*	*HAMP*	*MYBPC3*	*PPP1R13L*	*SLC40A1*	*TTR*
*DSC2*	*HFE*	*MYH6*	*PRKAG2*	*TAZ*	*VCL*

**Table 2 genes-11-01472-t002:** Demographics of the index-patients, genetic variants found and their families. ASH = asymmetric septal hypertrophy, AS = aortic stenosis, DCM = dilated cardiomyopathy, FS = fractional shortening, G + P- = genotype positive and phenotype negative, Htx = Heart transplantation, LVH = left ventricular hypertrophy, LVNC = Left ventricular non-compaction, LVOTO = left ventricular outflow tract obstruction, CM-gene panel = Next generation sequencing of 60 genes associated with SCD and cardiomyopathy, mVSD = muscular ventricular septal defect, RCM = restrictive cardiomyopathy, SCD = sudden cardiac death, Trio-exome = clinical exome (6794 genes with known clinical significance) in index patients and parents, ”trios”), VUS = variant of uncertain significance.

Family	Pedigree Annotation	Age at Diag.	Gender	HCM/CM SCD, ACA or Htx in Relative (n)	Phenotype of Index Patient	HCM/CM Individuals in Family	Family Members at Risk for HCM/CM	Family History	Genetic Results	ACMG Rules Applied and Classification/Associated with Disease in the Family
1.	II:1	7	F	2	ASH, FS 32%	5	11	Father Htx, Uncle SCD due to HCM, Cousin and cousin’s child with HCM	*MYH7* Chr14(GRCh37):g.23900677C>T, NM_000257.3:c.746G>A, p.R249Q	PS2 + PS4 + PM2 + PP1 + PP3 + PP5 Pathogenic/Yes
2.	I:1	22	M	1	Apical LVH	2	8	Mother SCD due to HCM	No result	No result
3.	II:1	14	M	0	First symptom pro-longed QTc, rapid evolving to HCM ASH and LVNC.	2	3	Mother mVSD, LVNC	*ABCC9* Chr12(GRCh37):g.21997457A>C, NM_005691.3:c.3275T>G p.I1092S	PP3 VUS/Yes
4.	II:4	8	M	2	ASH diastolic impairment	6	9	Sister, Father, Uncle with HCM, Uncle and grandfather SCD due to HCM	*FLNC* Chr7(GRCh37):g.128496973C>T, NM_001458.4:c.7559C>T, p.T2520I	PM2 VUS/Yes
5.	II:1	0.1	M	0	Severe LVOTO infant heart surgery	2	3	Father with pediatric onset of HCM	*MYBPC3* Chr11(GRCh37):g.47359334C>T, NM_000256.3:c.2320G>A, p.A774T	None VUS/Yes
6.	II:6	11	F	2	HCM/RCM	3	13	Index patient and brother with SCD as first symptom	*PGM1* Chr1(GRCh37):g.64100506G>A, NM_002633.2:c.689G>A, p.G230E	PM2 + PP3 + PS3 Pathogenic/Yes
7.	II:1	12	M	0	ASH and AS LVOTO	2	3	Father with HCM ASH	No result	No result
8.	III:1	12	M	2	ASH severe LVH	3	3	Mother and grandmother SCD due to HCM	*MYH7* Chr14(GRCh37):g.23899059C>T, NM_000257.3:c.1063G>A, p.A355T	PM2 + PP1 + PP3 + PP5 Likely Pathogenic/Yes
9.	III:1	15	M	0	LVNC/DCM	5	7	Mother and aunt with LVNC/DCM	*RBM20* Chr10(GRCh37):g.112572067C>G, NM_001134363.2:c.1912C>G, p.P638A and *VCL* Chr10(GRCh37):g.75873961C>T, NM_014000.2:c.2969 C>T p.A990V	PM1 + PP3 + PM2 + PM5 Likely Pathogenic/Yes PP3 VUS/Yes
10.	IV:1	24	M	1	ASH severe LVH	3	7	Mother with HCM, grandfather SCD and HCM, brother G+P-	*ALPK3* Chr15(GRCh37):g.85370829del, NM_020778.4(ALPK3):c.903del, p.(Ile301Metfs*10)	PVS1 + PM2 Likely Pathogenic/Yes

**Table 3 genes-11-01472-t003:** Assessment of the pathogenicity of the genetic variants in public online databases and pathogenicity prediction tools. * SIFT: D = Deleterious, T = Tolerated; MT (Mutation Taster): DC = Disease Causing, P = Polymorphism; PP (PolyPhen-2): PoD = Possibly Damaging, PrD = Probably Damaging, B = Benign.

Index	Variant	Protein	ClinVar	HGMD	gnomAD	SIFT/MT/PP *	CADD Score
Fam 1	NM_000257.3(MYH7): c.746G>A	p.(Arg249Gln)	RCV000015144.29 (Pathogenic/Likely pathogenic **—Familial hypertrophic cardiomyopathy 1), RCV000853263.2 (Pathogenic **—Cardiomyopathy), RCV000158761.4 (Pathogenic **—not provided), RCV000229956.5 (Pathogenic **—Hypertrophic cardiomyopathy), RCV000762925.1 (Pathogenic *—Familial hypertrophic cardiomyopathy 1), RCV000617265.1 (Pathogenic *—Cardiovascular phenotype).	CM910268 (DM)	-	D/DC/PrD	24.7
Fam 3	NM_005691.3(ABCC9): c.3275T>G	p.(Ile1092Ser)	RCV001050293.1 (Uncertain significance *—Dilated cardiomyopathy 1O).	-	ALL:0.0060% AMR:0.0056% NFE:0.011% OTH:0.014%	D/DC/PrD	29.3
Fam 4	NM_001458.4(FLNC): c.7559C>T	p.(Thr2520Ile)	-	-	-	D/DC/B	25.4
Fam 5	NM_000256.3(MYBPC3): c.2320G>A	p.(Ala774Thr)	RCV000035486.3 (Uncertain significance *—not specified), RCV000148678.1 (Uncertain significance *—Primary familial hypertrophic cardiomyopathy), RCV000770335.2 (Uncertain significance **—Cardiomyopathy), RCV000766349.1 (Uncertain significance *—not provided), RCV000415709.1 (Uncertain significance—Familial hypertrophic cardiomyopathy 4), RCV001071057.1 (Uncertain significance *—Hypertrophic cardiomyopathy), RCV000415662.1 (Uncertain significance—Left ventricular noncompaction 10).	CM120502 (DM)	ALL:0.0013% NFE:0.0035%	T/P/-	14.3
Fam 6	NM_002633.2(PGM1): c.689G>A	p.(Gly230Glu)	-	CM1618217 (DM)	-	D/DC/PrD	25.4
Fam 8	NM_000257.3(MYH7): c.1063G>A	p.(Ala355Thr)	RCV000225736.3 (Likely pathogenic *—not provided), RCV000769462.1 (Likely pathogenic *—Cardiomyopathy), RCV000470458.6 (Pathogenic/Likely pathogenic **—Hypertrophic cardiomyopathy), RCV000624861.2 (Pathogenic/Likely pathogenic **—Primary familial hypertrophic cardiomyopathy), RCV000620233.1 (Likely pathogenic*—Cardiovascular phenotype).	CM031268 (DM)	-	D/DC/PrD	24
Fam 9:1	NM_001134363.2(RBM20): c.1912C>G	p.(Pro638Ala)	-		-	D/-/PoD	23.7
Fam 9:2	NM_014000.2(VCL): c.2969C>T	p.(Ala990Val)	RCV000243207.2 (Uncertain significance *—Cardiovascular phenotype), RCV000539097.5 (Uncertain significance **—Dilated cardiomyopathy 1W), RCV000786265.1 (Uncertain significance—not provided), RCV000515262.1 (Uncertain significance *—Dilated cardiomyopathy 1W), RCV000038820.4 (Uncertain significance**—not specified).	-	ALL:0.029% NFE:0.055% FIN:0.028% OTH:0.055%	D/DC/B	29
Fam 10	NM_020778.4(ALPK3): c.903del	p.(Ile301Metfs *10)	-	-	ALL:0.0024% NFE:0.0044% FIN:0.0046%	-	24.1

**Table 4 genes-11-01472-t004:** Genes associated with congenital disorder of glycosylation (CDG), used in the clinical exome sequencing. CDG may be a cause of metabolic cardiomyopathy in the young.

*ALG1*	*B4GALT7*	*DPAGT1*	*MGAT2*	*PIGN*	*SLC35C1*
*ALG3*	*CHST3*	*DPM1*	*MOGS*	*PIGO*	*SLC35D1*
*ALG6*	*CHST6*	*EXT1*	*MPDU1*	*PIGT*	*SRD5A3*
*ALG8*	*CHST14*	*EXT2*	*MPI*	*PIGV*	*SSR4*
*ALG9*	*CHSY1*	*FKRP*	*NGLY1*	*PMM2*	*ST3GAL3*
*ALG11*	*COG1*	*FKTN*	*PGAP2*	*POMGNT1*	*ST3GAL5*
*ALG12*	*COG4*	*GALNT3*	*PGAP3*	*POMT1*	*STT3A*
*ATP6V0A2*	*COG5*	*GFPT1*	*PGM1*	*POMT2*	*TMEM165*
*B3GALNT2*	*COG6*	*GMPPB*	*PGM3*	*RFT1*	*TUSC3*
*B3GALT6*	*COG7*	*GNE*	*PIGA*	*SEC23B*	
*B3GAT3*	*COG8*	*ISPD*	*PIGL*	*SLC35A1*	
*B4GALT1*	*DOLK*	*MAN1B1*	*PIGM*	*SLC35A2*

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
