# Peer review of "Hereditary Hypertrophic Cardiomyopathy in Children and Young Adults—The Value of Reevaluating and Expanding Gene Panel Analyses"

_genes, 2020, doi:10.3390/genes11121472_

Round 1

Reviewer 1 Report

Several points were well modified, however, critical issues were remained.

Q1. In this manuscript, 2 variants in the MYH7 gene and 1 variant in the MYBPC3 gene were identified. It seems not clear which genes were tested in the first screening in these patients because these genes were most popular in cardiomyopahies.

Please clarify why the variants in MYH7 and MYBPC3 genes were not identified in the first screening and describe in the manuscript.

Otherwise, the authors should describe the reason in the study limitations.

Q2. The authors should show the list of genes for first screening for individual patients.

R: Thank you for this suggestion, we have added the list of included genes of the initial analysis for the first included 6 patients in the study; MYH7, MYBPC3, MYL2, MYL3, TNNT2, TPM1, ACTC, TNNI3, CSRP3, TCAP and PLN , and by time additional genes have been added to the panel such as MYH6. For children it is important to cover metabolic HCM, such as Danon disease (LAMP2) and HCM due to PRKAG2. Of note, the “old” genetic test did not have full coverage of the genes and just analyzed the pathogenic variants described at that time, that explain why the genetic variant was not detected earlier in the patients listed in Q1. Several labs still do not report the variants of unknown significance (VUS).

The authors should describe these points in the study limitations.

Q9. This study included over 20-year-old patients.

Q 10. The title should change “Pediatric hereditary hypertrophic cardiomyopathy” into “Hereditary hypertrophic cardiomyopathy”

This study apparently included the adult patients, however the authors did not modify the title.

Author Response

Thank you for your comments. All changes are made in red font in the revised manuscript.

Q1 … which genes were tested in the first screening … why the variants in MYH7 and MYBPC3 were not identified.

Reply: We have now included the information on genes in the results section for each family. We do not know the reason why the MYH7 and MYBPC3 variants were missed. The initial test was performed 2007 and the MYH7 variant in family 1 was not reported as a pathogen at that time, and could be a reason why the lab didn´t report it. Another speculation could be a lack of coverage at the time for the initial analysis. Probably the same situation for the and the MYBPC3 variant (VUS) in family 5.

We have contacted the lab that performed two of these analyses but have not yet received an explanation. Documentation of which lab performed the third analysis is missing.

Q2 … show the list of genes for the first screening for individual patients … The authors should describe these points in the study limitation

Reply: We have now included a passus on this in the study limitations.

Q9 & Q10 … over 20-year-old patients …The title should change

Reply: We have now changed the title to “Hereditary hypertrophic cardiomyopathy in children and young adults – the value of reevaluating and expanding gene panel analysis”

Reviewer 2 Report

In this study, the authors re-analyse 10 cases of paediatric cardiomyopathy to see if expanded gene panels, exome sequencing and new evidence for variant pathogenicity can identify pathogenic variants in patients with previous negative genetic testing results. They find likely pathogenic variants in 5/10 of these cases, including an interesting case of autosomal dominant ALPK3 inheritance.

The main limitation of this paper is that, although the authors state they applied ACMG variant interpretation for each detected variant, they provide no information as to how this was done or what ACMG rules were applied to each variant to justify the P/LP/VUS calls. ACMG interpretation is a standard part of clinical genetics workflows now, so that information should be already available or, if not, it should be applied to this research report. The authors only mention non-specific deleteriousness algorithms (and detail these in Table 3) - these are not a sufficient basis to determine pathogenicity. Instead, the authors should look at each class of ACMG evidence (population frequency, segregation etc) and apply these accordingly. For examples of how these are applied to HCM genes, see PMIDs 29300372, 30696458.

The detection of variants in MYH7, MYBPC3 (and perhaps RBM20) that were not reported in previous testing is noted in the discussion. If these were previously detected but considered VUS, this highlights the importance of re-analysis as demonstrated in this study. If they were not detected, that would be more concerning from a clinical genetics perspective. The authors state the reasons are unknown - what information did they have regarding the previous testing? Were only P/LP variants reported or were VUS at the time also reported back? This information should be available. The use of other reference transcripts is an unconvincing explanation since only a single relevant transcript is known for both MYH7 and MYBPC3.

The ALPK3 heterozygous truncating variant is a very interesting finding, given that most reports for this gene are associated with recessive inheritance. The authors refer to this in the discussion, but should cite and discuss two recently published studies looking at dominant inheritance for ALPK3 variants - Herkert et al (PMID:32480058) and Cheawsamoot et al (PMID:33191771) that would provide further support for the pathogenicity of this variant in their family.

Although ABCC9 has indeed been implicated in DCM, ClinGen re-evaluation suggests there is little evidence to support this association - https://search.clinicalgenome.org/kb/genes/HGNC:60

As the authors demonstrate in this study, re-analysis of previously tested patients is an important task to increase diagnostic yields by incorporating recently published novel disease genes (like ALPK3) and new evidence for variant pathogenicity. However, the converse to this is also true - variants (or genes) previously suspected to be pathogenic may now not be considered as such, due to a re-evaluation of published gene-disease evidence (e.g. ClinGen) or the availability of population databases like gnomAD highlighting that variants are too common to be disease-causing. It would be interesting for the authors to re-analyse with contemporary ACMG guidelines the previous 40 pathogenic variants detected in their cohort (though this may be outside the scope of this study). This issue could also be addressed in the Discussion.

Minor points:

I am not sure about the use of the term "exome sequencing" here, considering only about one-third of genes were sequenced. Perhaps something like "broad clinical panel" would be more appropriate? Exome sequencing implies all potential coding variants were assessed which is not the case here.

Introduction - "an actionable disease causing genetic variant can be found in nearly two thirds of the patients". Based on recent studies, this is far too high an estimate for HCM overall. It may be true for familial or paediatric HCM where there is a family history of disease, but not for HCM in general. I would suggest amending this.

The FLNC variant is wrongly described as p.V774M in the text.

Author Response

Thank you for the comments to our manuscript reviewed in Genes, genes-1008663. All changes are made in red font in the revised manuscript.

  • 2 … state they applied ACMG variant interpretation for each detected variant, they provide no information as to how this was done or what ACMG rules were applied …

Reply: The ACMG rules applied for classification of the variants have now been added to table 2. We have also added some text and the very appropriate references you suggested in the discussion and also a passus in study limitations.

  • 3 … The detection of variants in MYH7, MYBPC3 (and perhaps RBM20) that were not reported in previous testing … what information did they have regarding the previous testing …

Reply: Obviously, this information is important and we have now added in the results section which genes were analyzed for each family. We have requested further information from the laboratory that performed 2 of the missed variant analyses but we have not received any answer. We have also added a comment on this in the study limitations section. We do not know the reason why the MYH7 and MYBPC3 variants were missed. The initial test was performed 2007 and the MYH7 variant in family 1 was not reported as a pathogen at that time, and could be a reason why the lab didn´t report it. Another speculation could be a lack of coverage at the time for the initial analysis. Probably the same situation for the MYBPC3 variant (VUS) in family 5. Documentation of which lab performed the third analysis is missing.

  • 4 … The ALPK3 heterozygous truncating variant …

Reply: Thank you! We have now added some text in the discussion and included the 2 references you suggested.

  • 5  Although ABCC9 has indeed been implicated in DCM, ClinGen re-evaluation suggests there is little evidence to support this association

Reply: We have added this information as well in the results section

  • 6 … the converse is also true …

Reply: Yes, we have discussed this and might do it, but we think it is outside the scope of the present paper. A comment has now been added in the study limitations.

Minor points:

--- exome sequencing appropriate here?

Reply:   Well, we agree but “Clinical exome sequencing” was used by Illumina when introducing the kit TrueSight One. We believe that this term has been generally accepted so we have kept it in this manuscript. To use “broad clinical panel” would probably be confusing.

--- Introduction – “an actionable disease causing

Reply: We have tried to amend this.

--- The FLNC variant

Reply: Corrected

Thank you for an excellent review.

Yours sincerely,

Dr Antheia Kissopoulou, MD PhD student

[email protected]

Round 2

Reviewer 2 Report

The authors have addressed all of the comments and suggestions in the reviews and I am happy to recommend acceptance of this paper. Congratulations on a very nice study. 

This manuscript is a resubmission of an earlier submission. The following is a list of the peer review reports and author responses from that submission.

Round 1

Reviewer 1 Report

The authors aimed to identify the missing genetic etiology of the “gene-negative” cases in a cohort of pediatric cadiomyopathies, and showed 8 of 10 cases with genetic variants.

It is an important issue and valuable for readers, however, several concerns are raised.

Major comments

In this manuscript, 2 variants in the MYH7 gene and 1 variant in the MYBPC3 gene were identified. It seems not clear which genes were tested in the first screening in these patients because these gene were most popular in cardiomyopahies.

The authors should show the list of genes for first screening for individual patients.

The authors showed the demographics of the patients including patients’ findings, family history, and genetic variants. They also described the details about the present illness of each patient in the text.

But it is difficult for readers to understand what was a same finding and what was a different finding.

To understand the findings of these patients, the authors should reorganize table 3, for example, adding symptom, ECG findings, the existence of LVOTO (if possible, pressure gradient), etc.

In addition, the authors should add the summarized sentences of these patients in the text.

The authors showed 8 genetic variants, however, there were no reasons why these variants were pathogenic or VUS. They should clarify it.

They should also show the prevalence of gnomAD and the result of in silico analysis, ClinVar, and HGMD for each variant in the text or table 2.

Table 2 may had better divided into 2 tables; clinical characteristics and analysis of genetic variants.

From this study, did the authors conclude any results of clinical characteristics or genetic variants?

They had better state any genotype-phenotype correlation in the manuscript.

Minor

This study included over 20-year-old patients.

The title should change “Pediatric hereditary hypertrophic cardiomyopathy” into “Hereditary hypertrophic cardiomyopathy”

In materials, additional 4 patients were recruited.

Were they performed genetic testing before enrolled?

In table 2, regarding patient #10, what “G+P” stands for ?

In figure 2, the type of cardiomyopathies should clarify for each patients.

Reviewer 2 Report

Interesting case series. However, I do not feel that the results are wholly supportive of the main concept of the article, so that trio-exome and whole genome sequencing strategies enable better understanding of HCM cases or SCD due to HCM/CM in pediatric patients.

Data are few, mutations reported were VUS in 3/10 (33%), none in 2 (20%).

Family 9 do not meet inclusion  (no HCM, no family history of juvenile Tx or SCD due to HCM/CM)